# Determining Cognitive Workload Using Physiological Measurements: Pupillometry and Heart-Rate Variability

**DOI:** 10.3390/s24062010

**Published:** 2024-03-21

**Authors:** Xinyue Ma, Radmehr Monfared, Rebecca Grant, Yee Mey Goh

**Affiliations:** School of Mechanical, Electrical and Manufacturing Engineering, Loughborough University, Leicestershire LE11 3TU, UKr.p.monfared@lboro.ac.uk (R.M.);

**Keywords:** cognitive workload, task performance, pupillometry, heart-rate variability, cognitive-workload index

## Abstract

The adoption of Industry 4.0 technologies in manufacturing systems has accelerated in recent years, with a shift towards understanding operators’ well-being and resilience within the context of creating a human-centric manufacturing environment. In addition to measuring physical workload, monitoring operators’ cognitive workload is becoming a key element in maintaining a healthy and high-performing working environment in future digitalized manufacturing systems. The current approaches to the measurement of cognitive workload may be inadequate when human operators are faced with a series of new digitalized technologies, where their impact on operators’ mental workload and performance needs to be better understood. Therefore, a new method for measuring and determining the cognitive workload is required. Here, we propose a new method for determining cognitive-workload indices in a human-centric environment. The approach provides a method to define and verify the relationships between the factors of task complexity, cognitive workload, operators’ level of expertise, and indirectly, the operator performance level in a highly digitalized manufacturing environment. Our strategy is tested in a series of experiments where operators perform assembly tasks on a Wankel Engine block. The physiological signals from heart-rate variability and pupillometry bio-markers of 17 operators were captured and analysed using eye-tracking and electrocardiogram sensors. The experimental results demonstrate statistically significant differences in both cardiac and pupillometry-based cognitive load indices across the four task complexity levels (rest, low, medium, and high). Notably, these developed indices also provide better indications of cognitive load responding to changes in complexity compared to other measures. Additionally, while experts appear to exhibit lower cognitive loads across all complexity levels, further analysis is required to confirm statistically significant differences. In conclusion, the results from both measurement sensors are found to be compatible and in support of the proposed new approach. Our strategy should be useful for designing and optimizing workplace environments based on the cognitive load experienced by operators.

## 1. Introduction

The growing complexity and digitalization of the manufacturing sector have driven a disruptive change that may impose varying cognitive workloads on operators [1]. This transformation has also led to a new emphasis on monitoring cognitive workloads among manufacturing operators. This is typically accomplished through the application of engineering principles, such as assessing and improving human–machine systems, operators in the control room, maritime operators, and pilot and vehicle drivers [2,3,4,5]. Cognitive workload refers to the mental effort needed to perform the task, and can be influenced by a range of factors, including attention and capacity of information processing, working memory, and decision making [6,7]. An international survey has shown that workload-related stress is detrimental to the occupational health of European workers. This high-stress status can lead to decreased work performance, despite recent improvements in working environments and health care [8].

The Industry 5.0 paradigm [9] is using new technologies to shift the industrial management objectives from focusing solely on efficiency and productivity to putting the well-being of operators at the centre of production and boosting productivity between human workers, machines, and robots, which leads to digital and sustainable transitions. This approach builds on the acceptance of the existing Industry 4.0 standards and encourages the transition to a sustainable, human-centric, and resilient industry. The Industry 4.0 approach introduced cloud-based data connectivity, advanced artificial intelligence analytics, human interaction with robotics, automation systems, smart sensors, and virtual and augmented reality digital technologies [10,11]. In the context of emerging digitalized work environments, operators are required to upskill or reskill to accommodate the digitally transformed manufacturing sector. These requirements may challenge the operator’s information-processing capabilities and occupy an increased amount of working memory. Therefore, some operators’ available attentional resources may be insufficient for executing high-complexity tasks in the new digitalized workplace.

Furthermore, as mass customization in manufacturing is increasing due to many customers desiring customized products, the variability of product parts is increased, and the time for new product iteration is shortened [12]. The mass customization workflow also clearly shortens the operators’ cycle time due to the challenging delivery timelines. Cycle time is the official time allocated to operators to complete a given job on the assembly line, and it is becoming more restrictive to prevent potential supply chain risks in the production of customized products. These trends lead to increased complexity on the assembly line, and new production workflows may require more mental workload for operators [13]. The new production pattern in manufacturing places higher demands on cognitive workload [14], especially in the transition to a digitized paradigm. A survey of engineers at Swedish companies found that assembly complexity causing poor ergonomics can lead to more errors and scrap [15]. The well-being of employees must therefore be emphasized by enterprises to avoid the risk of cognitive-overload situations; this could also improve cost saving and waste reduction in manufacturing production.

In this context, it is crucial to constantly and accurately monitor operators’ cognitive workload to ensure that operators are not burned out, with consequences for the long-term and optimal range of operator performance. Manufacturing managers also reported that cognitive ergonomics is important in preventing manufacturing quality errors and operators’ health and disease issues [16]. Cognitive workload is an important feature of cognitive ergonomics; the expended mental effort and the associated attentional resources result in the cognitive workload level. Evaluating operators’ cognitive workload is essential for managing their well-being, which will also benefit the overall efficiency of achieving the enterprise’s manufacturing production goal in a sustainable, resilient, and ergonomic approach.

Three general methods exist to measure cognitive workload. First, self-reported and performance inference [17,18] are the most widely used methods in ergonomics; these two methods are generally accepted and can be applied in most cases. Along with the recent development of smart sensors, some studies have used physiological measurement methods [3,17,19,20] in the field of applied ergonomics, which reflects the desire to obtain more objective and precise measures of cognitive workload.

Physiological measurements are increasingly gaining attention in cognitive workload research, where wearable devices can be used to measure cognitive workload without impacting task performance. A study conducted on operators in control rooms found that eye-tracking metrics and task performance are suitable for indicating mental workload levels in monitoring tasks. Specifically, a significant decrease in mean relative pupil size has been reported from difficult to medium and simple conditions [19]. In a study conducted on the role of pilots collaborating with unmanned aerial vehicles, the physiological features of ECG and eye-tracking (ET) sensors were applied in the designated scenarios. In this pilot operational context, the ECG and ET features showed a statistical difference between high and low mental-workload conditions. Also, ECG features alone, or a combination of EEG and ET features, reached approximately 75% accuracy within the intra-subject classification [3].

Several studies have investigated the cognitive workload of operators in assembly tasks. The individual alpha frequency from EEG and blink rate from electrooculography (EOG) were shown to discriminate the cognitive-load levels between medium, high, and overload in the context of puzzle-related assembly tasks [17]. A vision-based cognitive-load assessment was also developed with a stereo camera under the 3D-printed objects’ assembly task [20]. The mental effort mean score from their novel assessment system in three-minute intervals was confirmed by the correlation between the physiological heart-rate variability and assembly performance. This study suggests that measuring cognitive workloads with an online human motion pattern-assessment system could give potential success in improving cognitive ergonomics in manufacturing. Further, the assembly task of a LEGO car set was investigated with the addition of an “N-Back task” to introduce increasing cognitive task difficulty by increasing the number N [18]. The N-Back task is commonly used to measure working memory capacity in attention and cognition research settings. The defined scenarios reported greater muscle activity and higher self-reported workload with increased cognitive demand.

Several studies have included the expertise factor in attention and cognition research. A study was conducted to explore the cognitive ability differences between musicians and non-musicians [21]. The results showed significant variations in reasoning and verbal memory abilities between the two groups. Moreover, musicians’ long-term professional training positively impacted their verbal memory. Similarly, it has been suggested that athletes perform better in mental imagery skills than non-athletes [22]. More recently, electroencephalographic (EEG) sensors have been used to study the impact of expertise on audio-visual cuts [23]. Media professionals were found to be more effective in dealing with the loss of visual information, as demonstrated by their reduced blink rate after audio-visual cuts. According to these studies, it is possible that cognitive abilities like reasoning, working memory, and imagery skills, which the level of expertise can influence, may also have an impact on the way non-experts handle manufacturing tasks, potentially challenging them to manage complex tasks in comparison to experts.

However, based on the existing literature, it remains unclear how cognitive workload affects performance in the context of manufacturing engineering. This is particularly important due to the current rapid implementation of digitalized technologies within this manufacturing domain and its complex impact on the user’s cognitive workload. This article proposes using two sensors, eye-tracking and electrocardiogram, together with related biomarkers to objectively measure cognitive-workload variations and their impact. There is currently no study specifically in this area that compares the performance of these two sensors and the resulting cognitive-load indices with task performance. Furthermore, no prior studies have reported the simultaneous use of transitions from pupillometry and cardiac bio-markers to cognitive-workload indices. Existing averaged metric-based biomarkers can only provide a value within a defined assembly period, which may not accurately assess dynamic changes in cognitive workload by reflecting accumulated mental effort.

In this study, participants were asked to play the role of an assembly operator performing the three assembly tasks of the Wankel Engine scenario as quickly as possible. The objectives of the presented study are to (i) investigate the ability of physiological biomarkers, ET and ECG, to differentiate between rest, low, medium, and high cognitive workloads, in addition to assessing their interaction with personalized factor expertise; (ii) define pupillometry and cardiac cognitive-workload indices that better indicate and align with the levels of complexity and their interaction with expertise; (iii) evaluate the effectiveness of the two identified cognitive-workload indices in aligning with task performance across different levels of complexity and their interaction with expertise factors.

## 2. The Proposed Method

### 2.1. Conceptual Model

Cognitive workload is a widely utilized terminology in cognitive ergonomics. It comprises two cause-and-effect components with mental demands (i.e., imposed workload) and strain (i.e., the workload expression on individuals), according to the ISO10075 standard definition [24]. In this research, the cognitive workload is referred to as the individual cognitive-workload expression affected by external sources, as shown in Figure 1.

The proposed conceptual cognitive workload model in Figure 1 follows the theory of demands and strain; however, this theory has been challenged as too simplistic in a review article on mental workload in ergonomics to describe cognitive workload to some extent [25]. Additionally, the fixed resource model could compensate for this simplistic cognitive workload cause-and-effect theory, i.e., humans are assumed to have limited attentional resource capacity for processing information and completing the task, which is also reviewed [25]. Moreover, they reviewed how demands in the conceptual model are factors comprised of external impacts such as task complexity and enterprises’ performance criteria. As an illustration, mental demands are imposed on the attentional resource balance, resulting in cognitive-workload fluctuations. The attentional resource occupied by mental effort is shown in Figure 1 and mediates the relationship between task demands and cognitive workload.

However, the fixed-resource theory still needs to consider non-attentional factors that compensate for their drawbacks in the actual application. Personal characteristic differences such as expertise are considered in this study as non-attentional factors that can potentially mediate cognitive workload and result in various cognitive-workload expressions. As shown in Figure 1, expertise, a non-attentional factor theoretically confirmed to influence available attentional resources, can modulate information processing and cognitive task completion [26]. Therefore, expertise is a mediating factor that can influence attentional resources in the cognitive-workload framework.

Human task performance has been used to indicate workload over long periods. Cognitive overload can probably lead to overall operators’ performance degradation, and this may not be compensated through mental effort when there is only a limited attentional resource while under a high cognitive load [27]. The lasting effect of cognitive overload and associated performance degradation would be detrimental to operator motivation and enterprise objectives. The cause-and-effect factors of cognitive workload examined in this research are represented by the proposed conceptual cognitive workload framework in Figure 1. In this figure, task complexity, which is a component of the external factor of task demand, serves as the input variable. The mediator factor of expertise influences attentional resources, and the factor of attentional resources can influence cognitive workload. Thus, the same complexity imposed on individuals can result in varying cognitive workloads. Cognitive workload also influences task performance; individual cognitive workload and task performance are defined in this framework as operator-evaluation metrics and output variables.

To evaluate this framework, a set of experiments was designed in which operators assemble an engine block, consisting of a series of tasks with varying levels of complexity, as described in Section 2.3. Three different assembly stages were designed to elicit low, medium, and high cognitive-workload levels. The cognitive workload is measured using the physiological method through pupillometry and electrocardiography (ECG) bio-markers. These methods, alongside the experimental design, are described next.

### 2.2. Participants and Equipment

Twenty-five healthy adults (age mean: 29 ± 10, 16 males and 9 females) were enrolled in the study. The study’s protocol was approved by the University Ethical Advisory Committee (proposal ID:2021-5254-4696). Some of the participants’ data had to be excluded from the datasets, primarily due to the quality of the data extracted for some experiments. Specifically, participants wearing corrective lenses result in an excessive blink rate, and pupil dilation as a biomarker of cognitive workload is sensitive and cannot be replaced by the median in this designed experiment. The final datasets with reliable data quality included 6 highly skilled participants with at least one year of assembly experience, grouped as “experts” in this article. The remaining 11 participants, grouped as “non-experts”, reported an average of 25 (±45) h of assembly experience in total. The non-experts had an average self-reported skill score of 0.71 (±0.91) on a scale of −2 to 2. The small sample of expert participants is comparable to similar studies that rely on highly skilled expertise [28,29], to understand the cognitive and physiological demands of multi-tasking, brain electrical activity, eye movements, and heart rate were recorded from 7 participants who simultaneously performed complex tasks at two difficulty levels [28]. Similarly, prior research employing small samples, like a study with six non-experts and two experts in complex surgical training, has successfully investigated the role of expertise in stress, attention, and acceleration [30].

Participants were asked to wear the Tobii Pro Glasses 3 eye-tracking device, manufactured by Tobii, based in Stockholm, Sweden [31], to measure pupil diameter as an indicator of cognitive workload [25] during the assembly task (Figure 2). Tobii Pro Glasses 3 provides robust and accurate eye-tracking metrics for measuring cognitive workload in assembly-scenario settings. The glasses allow participants to move and interact naturally with the physical assembly tasks while recording the defined biomarkers [31]. The Tobii Pro lab software (Lab version 1.181 from Tobii Connect, accessed date 22 May 2022) was utilized to mark events to individually locate the occurrence of the three stages of assembly on the eye-tracking timeline recordings. The pupil-diameter-timeline recordings and additional logged events were exported from the Tobii Pro lab as Excel files for further pre-processing and statistical analysis [32].

Participants were also asked to wear a chest belt ECG Zephyr Bioharness 3 (manufactured by Zephyr Technology, Annapolis, MD, USA) [33], a mobile device, to measure heart-rate variability metrics to investigate cognitive workload. It can provide participants with natural interactions in assembly scenarios and acquire highly accurate and reliable non-invasive cardiac biomarkers. Raw ECG waveforms were obtained using the Zephyr Log Downloader Tool (version 2016-04-07 from Zephyr Performance System, accessed date 7 June 2021) [34] and exported as Excel files for further pre-processing and statistical analysis.

These two sensors were chosen over other methods due to the fact that they provide more reliable and valid biomarkers of cognitive workload, and the eye-tracking sensor can also record participants’ assembly behaviours during the experiments. The physiological metrics obtained from both sensors, i.e., pupillometry and cardiac biomarkers, were analysed and compared to support the research findings and insights.

### 2.3. Task

The participants were asked to assemble a Wankel Engine, as illustrated in Figure 2. The assembly task consisted of three stages, and the assembly process was explained to the participants. The assembly task sequence was provided during the assembly task with a photographic image instruction. Figure 2a shows the first assembly stage to align and place the output shaft into the engine body. Figure 2b aligns the housing to the rods of the engine body to push together. Figure 2c–e shows the third stage of assembling the rotor into the housing, the cover, and screws, respectively. The experiment setup is shown in Figure 2f, with the participant assembling the Wankel Engine components seated at the workstation. The components were presented on the workstation for participants to conduct the assembly task within convenient reach. 

**Figure 2 sensors-24-02010-f002:**
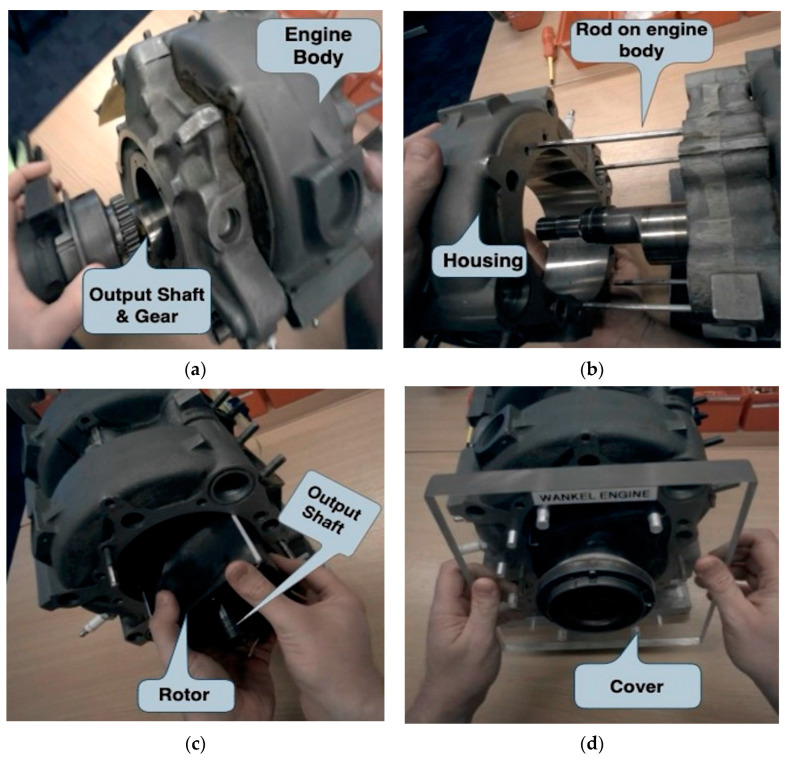
Three Wankel Engine assembly stages illustration and one participant assembled the engine at the working station. Three assembly stages are listed: (**a**) assembly stage 1: assemble output shaft and gear; (**b**) assembly stage 2: align rotor housing; (**c**) assembly stage 3: assemble rotor; as well as assemble cover and screws in (**d**,**e**). One participant assembled the engine components at the working station in (**f**), wearing eye-tracking and ECG sensors.

The task complexity of the three assembly stages was defined by a general theoretical model that allows three dimensions of task complexity: component complexity (TC1), coordinative complexity (TC2), and dynamic complexity (TC3) [35]. Dynamic complexity (TC3) is not involved in this experiment; it is typically present in the complex dynamic task, i.e., the air-traffic-controller task. Each stage of the engine assembly task only owns two types of task complexity: component (TC1) and coordinative (TC2).

In the first assembly stage of assembling the output shaft and stationary gear, the information cues are listed Table 1. The component complexity is assessed by calculating each required act’s accumulated information cues. There are two required acts in the first assembly stage; the pushing action is associated with the first information cue, and the rotating act is correlated with the second information cue. The coordinative complexity in the first assembly stage requires the pushing act to be done before the rotating act. The component complexity (TC1) for the first assembly stage is 2, and the coordinative complexity (TC2) is 1, and the task complexity equation with three different complexity dimensions can be shown as follows in (1): *TCt* = *αTC*_1_ + *βTC*_2_ + *γTC*_3_(1)
where *α* > *β* > *γ* and *α* = 2, *β* = 1, and *γ* = 0, the total complexity for the first assembly stage is 5.

In the assembly stage of assembling housing, the information cues are also listed in Table 1. There is only one required act aligning the holes on the housing with the four rods on the engine body. There are two information cues associated with this required act, resulting in the value of 2 for the component complexity; additionally, there is no sequencing between the required act, and consequently, the coordinative complexity is 0. As calculated from Equation (1), the total task complexity for the second assembly stage is 4. 

Additionally, the information cues of the third assembly stage are listed in Table 1; however, the acts are excluded if they are redundant with other acts. The act of alignment with information cues (3) and (4) in this third assembly stage is excluded in the aggregated distinct task acts due to appearing in the previous (housing) assembly stage. Therefore, the component complexity (TC1) is 3 instead of 5, accounting for the distinct information cues. The coordinative complexity requiring the assembling rotor has to be acted upon before aligning the cover and screwing the nuts in the third assembly stage. Consequently, the coordinative complexity (TC2) is 1 in this stage. As calculated from Equation (1), the total task complexity for the third assembly stage is 7. 

Table 2 illustrates the relationships between the assembly stage, task name, complexity levels, and scores. The experiment stage involving housing assembly, which has a score of 4 and is considered a low-complexity phase, can be performed in either the second or first stage. The stage involving gear, which has a score of 5 and is regarded as a medium-complexity stage, can be performed in either the first or second stage. Finally, the rotor- and cover-assembly stage, with a complexity score of 7 and rated as a high-complexity stage, is identified as the last stage. Therefore, the constructed task can be presented in a sequence from medium to low to high complexity, or from low to medium to high complexity. The order between the low- and medium-complexity stages was randomized to avoid any potential learning effects.

### 2.4. Experimental Protocol

Participants who reviewed and signed the informed consent form then completed a demographic survey (i.e., assembly experience level, health status, age, and gender) and were then fitted with the Tobii Pro Glasses 3 eye tracker and the Zephyr Bioharness 3 ECG wearable sensor. Once the eye-tracker device was calibrated to ensure sufficient pupillometry recording, the recording was started. Also, the ECG device was checked to work properly before the assembly tasks were launched. To minimize the impact of external factors on the experiment, the laboratory was kept quiet with nearly constant illumination and blackout blinds. 

The experiment started with a 2 min rest time before conducting the assembly tasks with open eyes, and this rest time was used to provide baseline signals. Participants were asked to complete three engine-assembly stages only once, in an assembly sequence of (1) or (2) gear assembly with output shaft and stationary gear, (2) or (1) housing, and (3) rotor and cover as quickly as possible. The paper-based instructions were presented to the participants at the start of the assembly task. The experimental session consisted of performing an assembly task for approximately 30 min, and the time required to complete the task depended on individual differences. 

### 2.5. Metric Extraction and Statistical Analysis 

#### 2.5.1. Pupil-Diameter Extraction 

The raw pupil-diameter metrics were collected at a sampling rate of 100 Hz and extracted for further pre-processing. The pupil-diameter-related parameters were extracted and pre-processed with Matlab_R2021b software [36]. Some invalid samples including artefacts were identified and removed, which were probably caused by blinks or system errors [37]. The invalid samples were rejected through quality-check measures and standard deviation filtering, and are considered as missing data. The quality-check measure is defined by the rule that a blink generally cannot be longer than 500 milliseconds; a blink exceeding 500 milliseconds is out of the interpolation criteria [38]. The outliers are detected and removed when the pupil-diameter values exceed the upper and lower boundaries of three standard deviations from the mean values. However, missing data can potentially reduce statistical power, and interpolation needs to be completed through either cubic-spline or linear interpolation [38]. A zero-phase low-pass filter with a cut-off frequency of 10 Hz smooths out and reduces noise spikes by removing high-frequency signals [37]. Baseline correction is performed individually after the pre-processing step by subtracting the mean pupil size during a “2 min rest period” before the start of the experiment [38]. Data quality was inspected through the calculation of the missing data percentage; in addition, this evaluation process was assisted by plotting. Trials with missing data percentages above 40% were excluded from the analysis due to the fact that interpolation for pupil-diameter data above 40% is not meaningful in this study, and these data may not be able to fully represent the fluctuations in cognitive workload that participants experienced during the assembly task, which can likely lead to misleading pupillometry results and interpretation. As a result, 5 participants were excluded based on this criterion.

Derivative pupil diameter is correlated to stress-related behaviours to some extent, and this metric is considered a potentially useful indicator of stress and possibly correlates with cognitive workload [39]. The pupil derivative was calculated as the pupil diameter difference between each two successive time samples; also, the averaged derivative was calculated in each assembly stage. In such a manner, the pupil diameter and derivative pupil-diameter biomarkers were extracted for further statistical analysis.

#### 2.5.2. Heart-Rate-Variability Extraction 

The raw ECG amplitudes were acquired at a 1 kHz sampling rate, and several cardiac parameters were extracted through defined pre-processing pipelines. The heart-rate variability parameters were extracted and pre-processed with Matlab_R2021b software [36]. This pipeline includes down-sampling the ECG signal to 250 Hz on account of improving computational efficiency, detecting R-R peaks using Pan–Tompkins’s algorithms [40], which includes DC bias elimination, signal normalization, low pass filter, high pass filter, derivative filter, squaring function, and moving window integration. Then, z-score correction is applied to correct a small number of errors, finding the outliers of an absolute z-score value larger than 2 and then replacing the outliers with the median value of R-R intervals. The Z-score is identified to measure how many standard deviations are below or above the population mean. We excluded 2 participants (who were also excluded from the pupillometry signals) based on the target heart-rate range for a healthy adult during physical activity, which is approximately 50% to 85% of the maximal heart rate, where the maximum is calculated as 220 minus age in years.

As a result of these signal-processing techniques, two pupillometry parameters and eight cardiac parameters of heart-rate variability (HRV) from 17 participants were extracted and summarized in Table 3. 

## 3. Results

### 3.1. Physiological Metrics 

The pupillometry and cardiac metrics obtained from the experiment are presented in Table 4, comparing the mean values between expert and non-expert groups in the four stages with three task complexities and one rest stage of the Wankel Engine assembly. Since most variables in Table 4 violated the normality, homogeneity of variance, and sample size assumptions of parametric statistical analysis, we opted for the Non-parametric Analysis of Longitudinal Data in Factorial Experiments (nparLD) technique [41], and these analyses were completed with R studio software (version 2021.09.2+382 from RStudio daily builts, accessed date 13 February 2022) [42]. A non-parametric statistical analysis of nparLD is applied on the two factors of assembly complexity and expertise, which aims to verify whether the physiological metrics have significant statistical differences between complexity or expertise or the interaction between complexity and expertise.

Two of these ten metrics showed statistical significance on the complexity factor and the promising ability to indicate cognitive workload; baselined pupil dilation (Wald type statistic of χ2(3) = 77.664, *p* < 0.001 ***; ANOVA-Type Statistic F(2.147) = 38.710, *p* < 0.001 ***) showed highly statistical significance, baselined PNN50 (Wald type statistic of χ2(3) = 9.650, *p* = 0.022 *; ANOVA-Type Statistic F(2.615) = 2.963, *p* = 0.038 *) showed statistical significance, and significant differences can be found in the interaction between complexity and expertise (Wald type statistic χ2(3) = 10.074, *p* = 0.018 *).

However, as shown in Table 4, other biomarkers lacked statistical significance for either the assembly stage or the expertise factor interaction. This might be due to a potential insensitivity of these metrics to workload differences within the predefined complexities or interactions with expertise groups. In the following paragraphs, we only report the details of baselined pupil dilation and PNN50 biomarkers, which showed statistically significant differences in the complexity or the interaction between complexity and expertise. Also, the changing trend of these two metrics in the low-, medium-, and high-complexity assembly stages between expertise groups will be interpreted and discussed. Furthermore, the tables summarize the results of non-parametric statistics, and the treatment effects for the parameters of baselined pupil dilation and PNN50 are presented in the Appendix A.

The mean results of baselined pupil dilation changes associated with the four stages of the engine assembly task are presented in Figure 3. The assembly stages are presented in the complexity sequence defined in Section 2.3, from low to high complexities of the housing assembly, gear assembly, and rotor- and cover-assembly stages.

Generally, as shown in Figure 3, the baselined pupil-dilation values are above the zero line in the three sub-engine-assembly stages. The multiple complexities defined in the engine-assembly stages can lead to increased cognitive-load arousal among the subjects; the non-experts showed higher metrics than the experts in the low- and high-complexity stages.

Regarding pupil-dilation means in Figure 3 and Table 4, in the non-expert group, a steady increase can be seen from low- to high-complexity tasks. The results indicate that the cognitive workload in the non-expert group aligns well with the complexity level in the sub-assembly stages. A marked increase in baselined pupil dilation in the expert group is shown from rest, low-, and medium-complexity stages. However, the misalignment between the complexity and pupillometry metrics among the expert group present in the high-complexity assembly stage; this might be because expertise can compensate for the high-complexity task, resulting in the pupillometry cognitive-load metric in high complexity being lower than the metric in the medium-complexity task.

Regarding measuring cognitive load with baselined pupil dilation (Appendix A), the non-experts showed higher relative treatment effects than the experts. The relative treatment effects (RTE) in both groups increased from low to medium complexity, ranging from 0.525 to 0.652 in experts and 0.630 to 0.671 in non-experts. The experts had a higher 95% confidence interval of RTEs than the non-experts, and this interval remained constant across different complexities for each group (Appendix A). When estimating cognitive load, the non-experts’ pupil dilation showed higher precision, while the experts’ pupil dilation showed higher variability than the non-experts.

The mean results of baselined PNN50 changes associated with task complexity are presented in Figure 4. Similarly, the assembly stages are presented in the complexity sequencing from low to high complexities. Non-experts’ PNN50 values are −0.044, 0.173 and −0.108, in low, medium and high complexities, respectively. Studies by other researchers have demonstrated a strong correlation between lower heart-rate-variability metrics and increased cognitive workload [43,44]. The values in low and high complexities are lower than the baseline value of zero, indicating a decreased PNN50 and higher cognitive workload in non-experts. Also, the cardiac metrics of non-experts are lower than the experts in each stage. Although the baselined PNN50 means did not demonstrate a higher cognitive workload in experts, according to Figure 4, the mean differences that are illustrated with mean dots and connected by line plots indicate a steady decrease from medium to high complexity. The experts exhibit a greater relative treatment effect (RTE) (0.673, 0.719, 0.684) than non-experts across levels of low to medium to high complexity (0.434, 0.516, 0.361) (Appendix A), suggesting that non-experts potentially have a higher cognitive workload than experts.

PNN50 values decreased from medium (gear) to high (rotor and cover) complexity in both experts and non-experts (Figure 4), but the pairwise comparisons between these observed values were not statistically significant after Bonferroni adjustment (Appendix A). Also, the lower HRV metrics from non-experts indicating higher cognitive workload can be found in the high-complexity task compared with the low or medium task. One possible explanation for non-experts demonstrating higher cognitive workloads in the low-complexity task is that the pre-defined difficulty level may have still triggered more significant cognitive arousal than the medium-complexity task for this group. The underlying reason for this is that performing the simple engine-assembly task still needs non-experts to occupy considerable attentional resources for information processing compared with medium complexity; this could also be interpreted by the low- and medium-complexity tasks’ scores being close to each other. As shown in Figure 3 and Figure 4, the baseline pupil dilation and PNN50 showed promising abilities with statistically significant differences to indicate cognitive workload in the three task complexities (low, medium, and high), and the differences can be seen between the two expertise groups. However, it remains difficult to compare the trends of the two cognitive workload metrics because the different biomarkers show opposite changes in indicating high cognitive workload.

Due to the shortcomings of existing methods, a novel cognitive-workload index conversion is proposed in the next Section 3.2 to support and further verify that the cognitive workload biomarkers from ECG and ET sensors can discriminate the task complexities through statistical analysis and the demonstration of cognitive-workload-trend graphs across different expertise groups.

### 3.2. Cognitive-Workload Indices

The primary reasons for converting cognitive-workload indices are to shorten the measurement interval and improve the accuracy and reliability of cognitive-workload estimation. This is obtained by averaging a few seconds of intervals during each assembly stage. Secondly, the cognitive-load indices also can improve the consistency for indicating high cognitive workloads by aligning fluctuations in the same direction rather than opposite directions from different sensors and biomarkers. The analysis technique of non-overlapping 6 s epoch averages of eye-tracking metrics was applied to the estimation of mental workload for the pilot’s cooperation task [3]. However, this metric-based interval method is an instantaneous indicator, and it is not suitable to be accumulated in a defined period for estimating the cognitive workload. The processing of two cognitive-load indices proposed by the authors can address this issue and will be illustrated in the following paragraphs.

Absolute peak dilation responses to baseline will be utilized to ascertain and convert them into the pupillometry cognitive-load index. Identifying peaks and averaging them within a short time window is an effective method for capturing elicited pupil dilation, as opposed to using mean amplitude, which eliminates the characteristics of high cognitive workload [45]. Figure 5 shows an example of the pupillometry cognitive-load index conversion. To measure cognitive workload, a technique called epoch analysis was applied. This technique involves dividing the pupil-dilation data into non-overlapping 7 s segments (epochs). For each epoch, the average peak pupil dilation was calculated. Peak pupil dilations greater than zero were converted into a high cognitive-workload index of 1, as previous research has shown that pupil diameter is positively correlated with cognitive workload. During epoch analysis, peak pupil dilation values less than zero were converted to low cognitive-workload indices of 0. This process was repeated until the last 7 s epoch was analysed. The high cognitive-workload indices were then accumulated to produce total high cognitive-workload indices for each assembly-complexity level. 

Figure 6 shows an example of the cardiac cognitive-load index conversion. In order to keep the similar measurement frequency and data length between the two different metrics, a non-overlapping 10 s means of cardiac metrics technique was applied. HRV signal analysis can be performed at ultra-short-term intervals of 10 s to 5 min; maintaining an HRV signals analysis window of at least 10 s is required to perform an optimal HRV assessment in the defined activities [46]. Moreover, in the assembly stage, mean baselined PNN50 interval values less than zero were converted to high cognitive-workload indices of 1, and mean baselined PNN50 interval values within each epoch greater than zero were converted to low cognitive-workload indices of 0, due to lower heart-rate variability values having been shown to correlate with higher cognitive workload. This process was repeated until the mean baselined PNN50 value for the last 10 s of the epoch was converted. The high cognitive-workload indices for each assembly complexity level were summed to produce total high cognitive-workload indices. The pupillometry and cardiac indices are presented in Table 5. 

The conversion processes of the two measurement methods have been completed from the original biomarkers to the cognitive-load indices. The cognitive-load indices are summarized in Table 5 with accumulated the cognitive-load indices of numerous segments (epochs) in each task complexity, namely rest (baseline), housing (low complexity), gear (medium complexity), and rotor and cover (high complexity) stages. The Pearson correlation coefficient in Table 5 reveals that the degree of correlation between the indices varies across different levels of complexity. The low-complexity (housing)-assembly stage exhibits a highly positive correlation with r = 0.826 (*p* < 0.001, α = 0.05), whereas the medium-complexity (gear)-assembly stage does not exhibit any correlation. The high-complexity (rotor and cover)-assembly stage shows a moderate positive correlation with r = 0.454 (*p* > 0.05, α = 0.05).

### 3.3. Statistical Analysis for Cognitive-Workload Indices 

The statistical analysis of pupillometry and cardiac cognitive-load indices showed highly significant differences between task-complexity stages. Highly significant differential effects were observed for both pupillometry and cardiac indices across conditions (*p* < 0.001), as evidenced by both ANOVA (F(2.413) = 35.335, F(2.507) = 9.555) and Wald-type statistics (χ2(3) = 280.264, χ2(3) = 31.645). However, none of them showed statistically significant differences between the interaction of assembly complexity and expertise factors. Statistical differences in cognitive-load indices between assembly stage and expertise factors were investigated correspondingly with the Non-parametric Analysis of Longitudinal Data in Factorial Experiments [41]. The Appendix A contains tables and figures that summarize and illustrate the results of non-parametric statistical tests and the treatment effects on pupillometry and cardiac cognitive-load indices.

As shown in Figure 7, the pupillometry cognitive-load indices of assembly stages are higher than the resting stage. At the same time, the high-complexity cognitive-load indices are significantly higher than the indices in the low- and medium-complexity stages. Although there is no significant difference between complexity and expertise factors, the non-experts from low and high complexities had higher cognitive loads than the experts. The non-experts’ higher pupillometry index can also be seen from relative treatment effects that non-experts own higher RTEs (0.577, 0.860) than experts (0.429, 0.777) in low and high task complexities (see Appendix A). With regard to non-experts’ cognitive-load indices, the increased trend can be seen in the comparisons from low to high and medium to high. The experts’ cognitive-load indices remained constant from low- to medium-complexity tasks. At the same time, markedly increased indices from medium to high complexity are shown in experts.

As illustrated in Figure 8, the non-experts’ cardiac cognitive-load index is higher than the experts’ cardiac cognitive-load index in each complexity task. The experts’ cardiac cognitive-workload showed a steady increase from low-complexity- to the high-complexity-assembly stage. In terms of relative treatment effects (see Appendix A), non-experts displayed relatively higher values (0.548 and 0.804) in low and high task complexities compared to experts (0.279 and 0.673). In the medium complexity, non-experts exhibited a slightly higher relative treatment (0.316) than the experts (0.306). An increase in cardiac cognitive workload was observed for both experts and non-experts as task complexity within their respective groups increased. The exception was non-experts at medium complexity, who exhibited lower cardiac workloads compared to the low complexity level.

### 3.4. Task Performance 

As shown in Figure 9, overall task completion time increases with the task complexity from low to high in the two expertise groups, except for non-experts in the medium-complexity stage. The statistical analysis results show that the task-complexity factor significantly affected assembly performance with a *p*-value < 0.001 in Wald-type statistics and ANOVA-type statistics. However, no statistically significant differences were shown in the interaction of complexity and expertise. The increase in assembly time was identified in both expert and non-expert groups when comparing the high complexity level with the low or medium complexity level. Non-experts have a relatively higher relative treatment effect (RTE) (0.588 and 0.862) compared to the experts’ RTE (0.480 and 0.779) in the low and high complexities (see Appendix A). However, the medium task complexity showed that experts (0.505) have slightly higher RTEs than the non-experts (0.486). 

## 4. Discussion

Our approach proposes that the physiological biomarkers of pupil dilation and cardiac PNN50 can be used to estimate cognitive workloads, depending on assembly complexity and expertise. The results showed that these two biomarkers had statistically significant differences depending on the assembly-task complexity, with the cardiac biomarker of PNN50 having statistically significant differences in the interaction between complexity and expertise within this task.

Notably, we found that averaging the cognitive workload experienced during each assembly stage did not align well with the three complexity levels, nor with the cumulative and dynamic cognitive workload encountered during the assembly process. Additionally, evaluating physiological biomarkers may not be a fair comparison method due to the unique variations in pupillometry and cardiac metrics indicating a high cognitive workload. Therefore, cognitive-workload indices were introduced to capture the dynamic and accumulated cognitive loads and to conserve the consistency of high-cognitive-workload changes between pupillometry and cardiac biomarkers. The pupillometry and cardiac cognitive-workload indices were verified as statistically significant in the assembly complexity. Our findings align with previous research on cognitive-workload-indicator development. In studies on assembly tasks [20], novel cardiac indices derived from heart-rate variability demonstrated similar trends to assembly performance across varying complexity levels within short intervals. Additionally, a pilot role study [3] employing multiple physiological sensors (ECG and ET) found statistically significant differences in features between high- and low-mental-workload conditions. Additionally, in our study, the cognitive-workload indices derived from these two sensors also demonstrate clear responsiveness to complexity levels (rest, low, medium, and high), further confirming the development of effective biomarkers. This finding is similar to that of the study examined, in which increased cognitive demand for assembly tasks was reported as a higher workload [18]. However, this phenomenon was only observed in this study when comparing low- and high-complexity conditions and medium- and high-complexity conditions. Similarly, another study found that individual alpha frequencies from EEG and blink rate from EOG can discriminate between medium, high, and overloaded cognitive-load levels in the puzzle-solving assembly task [17]. However, in this study, the cognitive-workload indices could not distinguish complexity well between low and medium conditions. This is due to Morton and colleagues introducing the additional three different working memory tasks with each complexity level to increase the distinction between the three complexity levels within the task. Utilizing pupillometry and cardiac indices in this study enhances our understanding of how cognitive workload responds to complexity in the assembly task. Their capacity to visually illustrate and compare these changes enhances a better recognition and understanding of cognitive load dynamics.

The pupillometry index and the cardiac index showed a similar trend that experts had lower cognitive workloads than non-experts in all the assembly complexities (Figure 7 and Figure 8), except for pupillometry indices in medium complexity, although statistically significant differences were not exhibited in the interaction of complexity and expertise. This observation can be considered a verification that personal factor expertise potentially exerts an influence on cognitive workload within an industrial experiment setting, and may need to use more specified scenarios to verify the statistically significant differences between complexity and expertise.

The non-expert group showed higher cognitive-load indices in the pupillometry and cardiac indices graphs when comparing low and high complexity and medium and high complexity. Experts showed an increasing trend in cognitive-load indices for pupillometry and cardiac signals from low to high complexity, with a steady increase or constant variation between low and medium complexity. Furthermore, cognitive overload can be found in non-experts compared with experts in the low and high complexities, as shown from both cognitive-load indices. The lack of alignment between low or medium cognitive-load indices and task complexity may be due to the fact that the difference in complexity scores is not distinctive. Nonetheless, significant increases in cognitive workloads can be found in the high-complexity assembly task compared with the lower-complexity tasks.

The two cognitive-workload indices were correlated in the low- and high-complexity conditions, but not in the medium-complexity condition. As shown in Table 5, the cardiac index showed exceptionally low scores of 0 for three participants in the medium-complexity condition, compared to the pupillometry index. This finding is likely attributed to the fact that the medium-complexity task was the first task for them and that anxious participants did not exhibit substantial changes in their HRV from the baseline to the early stage of stressors [47]. However, in highly complex tasks, such as multiple varied-assembly tasks, cognitive-workload indices are better indicators of cognitive load than original physiological biomarkers, as physiological biomarkers may not clearly show the relationship between task complexities and cognitive loads.

We found that the eye-tracking device is ideal when measuring cognitive workload in assembly scenarios, as high cognitive-workload results can be traced back to specific assembly segments in the recording video. Also, such signals are less likely to be influenced by psychological factors like anxiety.

The performance metric showed statistical significance in the assembly complexity, and performance can also indicate that higher cognitive workloads inevitably lead to performance degradation, such as longer completion times [25]. The most notable finding is that assembly completion time was strongly correlated with the two cognitive-workload indices mentioned above. The same trends in experts and non-experts can even be found by comparing the performance plot Figure 9 and the pupillometry cognitive-load index plot Figure 7.

The existing method for estimating cognitive workloads with averaged pupil-dilation biomarkers could not capture the accumulated cognitive workload of short intervals. As shown in Figure 3, experts exhibited a slightly higher cognitive workload in the high complexity task than in the low complexity task. When compared with existing pupillometry biomarkers, the novel proposed method in the pupillometry cognitive-workload index, which uses short-interval estimation, enables the accumulation and capture of cognitive workloads in the high complexity task in both groups. This, in turn, increases the cognitive-load distinction between low and high and medium and high complexities (Figure 3 and Figure 7).

The existing method of estimating cognitive workload using averaged PNN50 measures results in similar fluctuations to the cardiac cognitive-workload index, but with the opposite indication of high cognitive workload. Using original multiple biomarkers to indicate cognitive load in this context can be misleading, because fluctuating biomarkers with different directions can mask the true variation in cognitive workload across complex tasks. The proposed method, which utilizes the cardiac cognitive-workload index, reverses the direction of the indication to align with the complexity level. This provides better discrimination between non-experts in low- and high-complexity tasks, as well as between experts across all three complexity levels, as shown in Figure 4 and Figure 8.

This new approach may still not be good at distinguishing the cognitive workload between low- and medium-complexity tasks, especially in non-experts. This is potentially due to the bio-sensors being insufficiently sensitive to distinguish the cognitive workload between these two tasks. Another underlying reason may be that the non-experts may have difficulties performing the medium-complexity task of gear assembly to the right standard, which can considerably decrease the cognitive workload in the medium-complexity task. Due to the engine-assembly sequence, the first two stages (low and medium complexities) must be completed successfully to proceed to the high complexity (rotor and cover) assembly, and the errors in the earlier stage may largely influence the last stage’s task by increasing cognitive workload and completion time.

The comparison of the two cognitive-load indices shows that the cardiac cognitive-load index is better at indicating cognitive workload in the assembly task scenario, with a better alignment between low, medium, and high complexity levels for experts. It is also more likely to differentiate cognitive workload between experts and non-experts at the same task-complexity level. The pupillometry cognitive-load index is well aligned with task performance in terms of completion time, especially from low- to high-complexity assembly stages, in both experts and non-experts.

There are two limitations to this study. Firstly, there is a lack of perceived workload to compare the objective data with participants at different stages of the task. Future research should include self-reported workloads at short intervals within the defined production scenario to gain a broader understanding. Secondly, the relatively low statistical power of 0.3 could potentially limit the accuracy of the proposed model and the detection of the effects. However, considering that the effect size of approximately 0.6 is within the acceptable range for engineering psychology research, this study still provides valuable insights into cognitive-workload variations in response to task demands. Future studies should address the power limitation by potentially increasing the sample size.

## 5. Conclusions

Our results highlight the importance of understanding operators’ cognitive workload in a human-centric digitalized manufacturing system. We introduce a new approach to analyse bio-markers from wearable sensors for eye-tracking and ECG to indicate cognitive workload and compare this with existing methods. The relationship between the cognitive workload, task complexity, and the expertise level of the operators was also determined. Specifically, a new workload-index conversion method was introduced and proved to be effective when participants are completing the different levels of complex tasks. Both pupillometry and cardiac cognitive-load indices, along with completion time performance, exhibit consistent results. Notably, the two indices align well in indicating cognitive workload across different complexities, reflecting similar trends to completion time in both experts and non-experts.

The proposed hypothesis was tested and supported by a practical experiment, where participants went through a series of assembly tasks with defined complexity levels. Furthermore, the expertise factor was uniquely used in this research as a mediator to indicate a more accurate method of measuring the impact of task complexity on cognitive workload and, therefore, the performance of the operators. This was found to be particularly relevant within the engineering sector of manufacturing assembly. However, the two cognitive-load indices did not reveal any statistically significant differences in terms of expertise or the interaction between complexity and expertise across the tasks. 

Prior studies have shown that expertise has a significant impact on cognitive and attentional performance in other fields, such as music, media, and sports [21,22,23]. To validate the assumption of expertise differences in manufacturing assembly tasks, more specialized scenarios need to be designed and conducted.

Future research should focus on the impact of additional personal traits on variations in cognitive workload in manufacturing settings, where operators interact with advanced digital technologies and interfaces.

## Figures and Tables

**Figure 1 sensors-24-02010-f001:**
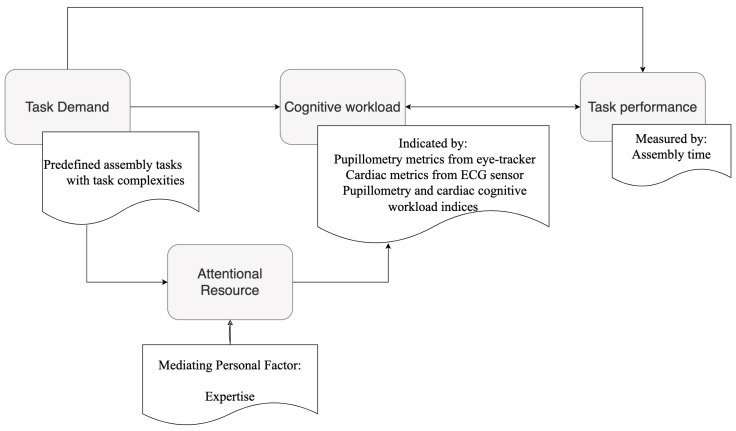
Conceptual cognitive workload framework.

**Figure 3 sensors-24-02010-f003:**
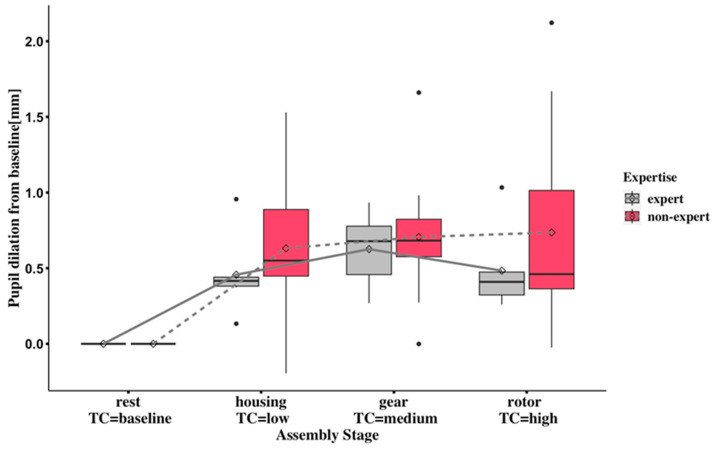
Comparison of baselined pupil dilation for two expertise groups under rest (baseline), housing stage (low complexity), gear (medium complexity), and rotor (high complexity). The black circles outside the boxplots represent extreme values that indicate high (high value) or low (low value) cognitive workloads compared to the overall pattern of cognitive workloads.

**Figure 4 sensors-24-02010-f004:**
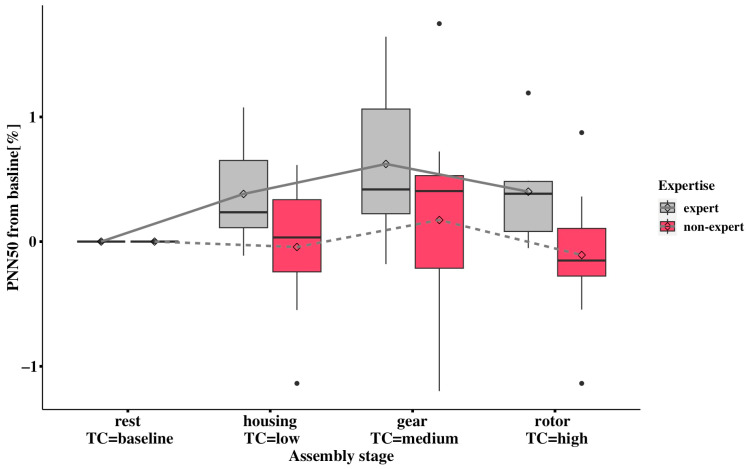
Comparison of baselined PNN50 for two expertise groups under rest (baseline), housing (low complexity), gear (medium complexity), and rotor (high complexity). The black circles outside the boxplots represent extreme values that indicate high (low value) or low (high value) cognitive workloads compared to the overall pattern of cognitive workloads.

**Figure 5 sensors-24-02010-f005:**
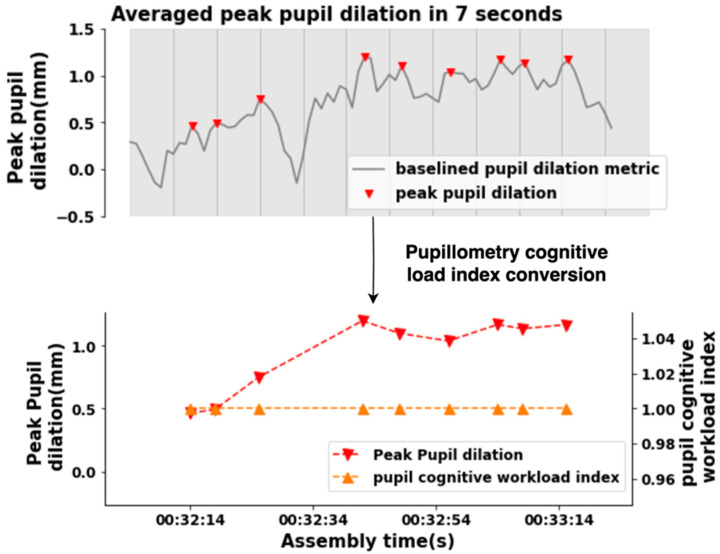
Pupillometry cognitive-load index converted from baselined pupil-dilation metrics for subject 105.

**Figure 6 sensors-24-02010-f006:**
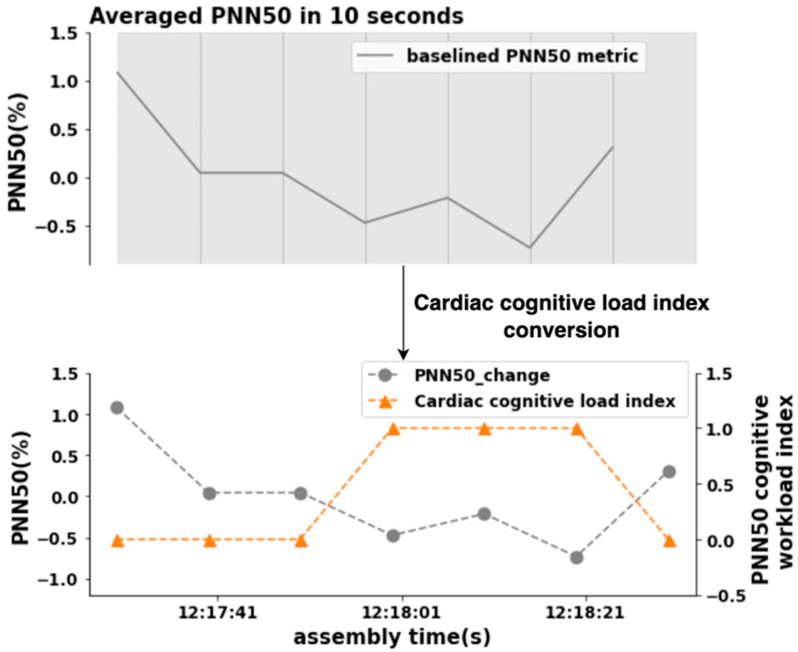
Cardiac cognitive load index converted from baselined PNN50 metrics for subject 107.

**Figure 7 sensors-24-02010-f007:**
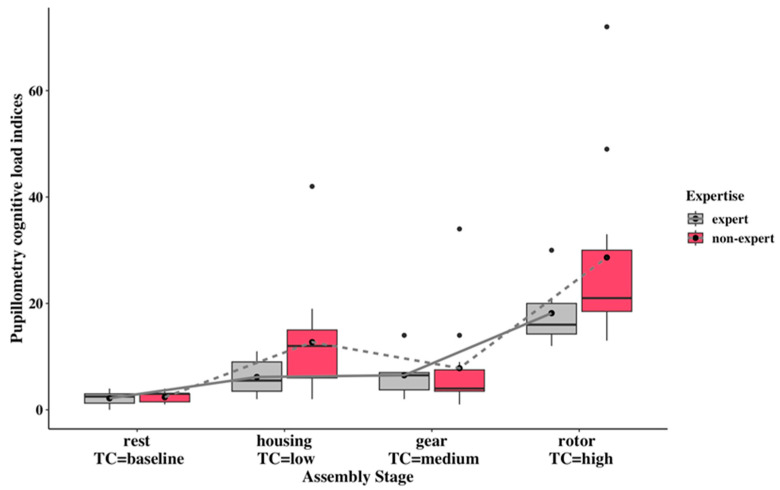
Comparison of pupillometry cognitive-load index for two expertise groups under multiple stages of rest (baseline), housing stage (low complexity), gear (medium complexity), and rotor (high complexity). The black circles above the boxplots represent extreme values that indicate high (high value) cognitive workloads compared to the overall pattern of cognitive workloads.

**Figure 8 sensors-24-02010-f008:**
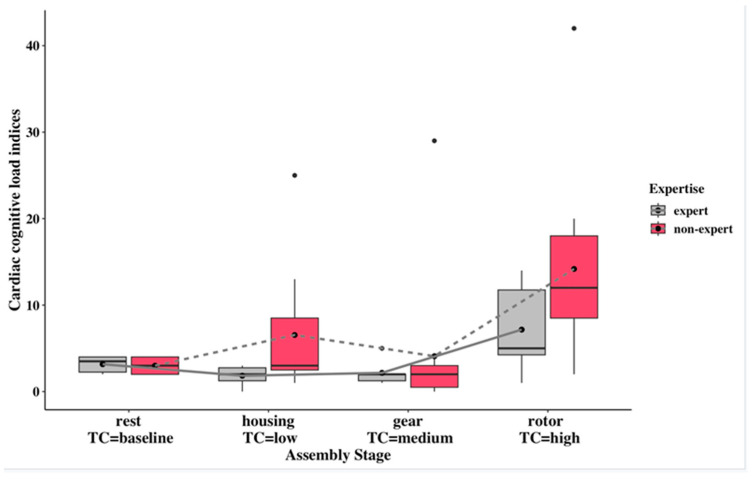
Comparison of cardiac cognitive-load index for two expertise groups under multiple stages of rest (baseline), housing (low complexity), gear (medium complexity), and rotor (high complexity). The black circles above the boxplots represent extreme values that indicate high cognitive workloads (high value) compared to the overall pattern of cognitive workloads.

**Figure 9 sensors-24-02010-f009:**
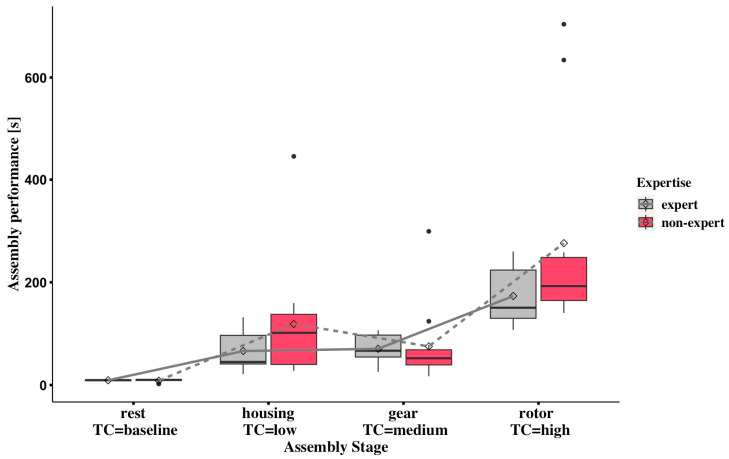
Comparison of completion time for two expertise groups under multiple stages of rest (baseline), housing (low complexity), gear (medium complexity), and rotor (high complexity). The black circles above the box plots represent extreme values indicating longer completion time (high value) compared to the overall pattern of completion time.

**Table 1 sensors-24-02010-t001:** Required acts and information cues for Wankel Engine assembly.

Assembly Stage	Required Acts	Information Cues
Gear assembly: first (or second) assembly stage	1. Pushing (behaviour act 1) and rotating (behaviour act 2) the output shaft into the hole of the engine body.	1. The correct side of the hole for output assembly;
		2. Quality of assembly completion of the gear of the output shaft.
Housing assembly: second (or first) assembly stage	2. Align (behaviour act 1) the 4 holes on the housing with the four rods on the engine body.	1. Visualization of 4 rods of the engine;
		2. Visualization of 4 correlated holes of engine rods on the housing.
Rotor and cover assembly: third assembly stage	3. Assemble (behaviour act 1) the rotor inside the housing, align the correct holes on the cover with four rods on the engine body, and screw (behaviour 2) 5 nuts.	1.Visualization of the output shaft, housing chamber, and rotor;
		2. Quality of assembly completion for the rotor;
		3. Visualization of 4 rods on the engine body;
		4. Visualization of 4 correlated holes on the cover;
		5. Finding 5 nuts.

**Table 2 sensors-24-02010-t002:** Assembly stage, assembly task name, task complexity levels, and score illustration.

Assembly Stage	Assembly Task Name	Task Complexity	Task Complexity Score
2 or 1	Housing assembly	Low complexity	4
1 or 2	Gear assembly	Medium complexity	5
3	Rotor and cover assembly	High complexity	7

**Table 3 sensors-24-02010-t003:** Cognitive-workload metrics overview.

Cognitive Workload Metrics	Definition
Baselined pupil diameter	The average pupil dilation from baseline time was subtracted from the pupil diameter.
Baselined pupil-diameter derivative	The average pupil-diameter derivative from baseline time was subtracted from the pupil-diameter derivative, and this metric can quantify the extent of pupil dilation or constriction from baseline time.
Baselined standard deviation of the RR intervals (baselined SDNN)	The average SDNN from baseline time was subtracted from the standard deviation of the RR intervals (SDNN).
Baselined Root Mean Square of successive differences between normal heartbeats (baselined RMSSD)	The average RMSSD from baseline time was subtracted from the Root Mean Square of successive differences between normal heartbeats (RMSSD).
Baselined proportion of the number of pairs of successive NN intervals that differ by more than 50 ms divided by the total number of NN intervals (baselined PNN50)	The average PNN50 from baseline time was subtracted from the proportion of the number of pairs of successive NN intervals that differ by more than 50 ms divided by the total number of NN intervals (PNN50).
The low-frequency band (LF)	The low-frequency band is from 0.04 to 0.15 Hz.
The high-frequency band (HF)	The high-frequency band is from 0.15 to 0.4 Hz.
The normalized low-frequency band power (LFnu)	The normalized low-frequency band power is from 0.04 to 0.15 Hz.
The normalized high-frequency band power (HFnu)	The normalized high-frequency band power is from 0.15 to 0.4 Hz.
The ratio of low-frequency to high-frequency (LF/HF ratio)	The ratio of low-frequency to high-frequency is LH/HF.

**Table 4 sensors-24-02010-t004:** Mean and standard deviation results of analysis of physiological features in engine assembly.

Physiological Metrics	Expertise Group ^1^	Rest (B) ^2^	Housing (LC) ^2^	Gear (MC) ^2^	Rotor and Cover (HC) ^2^
Baselined peak pupil dilation (mm) ^3^	E	0.000 ± 0.000	0.457 ± 0.270	0.626 ± 0.253	0.484 ± 0.282
	N-E	0.000 ± 0.000	0.631 ± 0.469	0.706 ± 0.419	0.736 ± 0.682
Baselined pupil- diameter derivative (mm/s) ^3^	E	0.000 ± 0.000	6.993 ± 0.114	−0.041 ± 0.152	0.015 ± 0.026
	N-E	0.000 ± 0.000	−0.011 ± 0.010	−0.002 ± 0.032	−0.001 ± 0.006
Baselined SDNN (ms) ^3^	E	0.000 ± 0.000	4.363 ± 22.123	−4.257 ± 13.977	5.780 ± 18.861
	N-E	0.000 ± 0.000	30.434 ± 61.439	−13.292 ± 33.977	−2.375 ± 58.385
Baselined RMSSD (ms) ^3^	E	0.000 ± 0.000	18.934 ± 30.241	9.051 ± 18.609	16.374 ± 27.140
	N-E	0.000 ± 0.000	−34.743 ± 86.464	−8.940 ± 47.154	1.340 ± 75.709
Baselined PNN50 (%)	E	0.000 ± 0.000	0.382 ± 0.454	0.623 ± 0.686	0.401 ± 0.449
	N-E	0.000 ± 0.000	−0.044 ± 0.501	0.173 ± 0.787	−0.108 ± 0.510
LF (ms^2^/Hz) ^3^	E	1.196 ± 2.825	14.776 ± 22.135	12.411 ± 26.253	0.909 ± 1.432
	N-E	8.577 ± 28.084	0.797 ± 1.150	4.760 ± 13.400	1070.000 ± 34,400.000
HF (ms^2^/Hz) ^3^	E	0.021 ± 0.049	0.176 ± 0.274	0.147 ± 0.300	0.008 ± 0.012
	N-E	0.104 ± 0.344	0.011 ± 0.021	0.012 ± 0.024	1770.000 ± 5860.000
LFnu (%)	E	0.425 ± 0.364	99.505 ± 0.554	98.923 ± 0.606	99.263 ± 0.568
	N-E	0.382 ± 0.506	82.252 ± 37.974	90.390 ± 29.900	81.900 ± 35.703
HFnu (%)	E	0.005 ± 0.008	0.495 ± 0.554	1.077 ± 0.606	0.737 ± 0.568
	N-E	0.003 ± 0.007	0.562 ± 0.675	0.539 ± 0.857	2.345 ± 4.320
LF/HF ratio (Unitless)	E	425.491 ± 283.706	616.648 ± 541.243	179.885 ± 215.633	281.295 ± 258.100
	N-E	566.180 ± 177.973	408.872 ± 363.832	527.458 ± 423.384	400.036 ± 729.143

^1^ Column Expertise Group is presented with E for experts and N-E for non-experts. ^2^ Rest: baseline stage; housing assembly: low-complexity stage; gear assembly: medium-complexity stage; rotor and cover assembly: high-complexity stage.^3^ Units: “mm” is an abbreviation for millimetres; “mm/s” stands for millimetres per second; “ms” stands for milliseconds, “ms^2^/Hz” stands for milliseconds squared per hertz.

**Table 5 sensors-24-02010-t005:** Pupillometry and cardiac cognitive-load indices.

Pupillometry Cognitive-Load Indexes					Cardiac Cognitive-Load Indexes			
Subject ID	Rest (B)	Housing (L)	Gear (M)	Rotor and Cover (H)	Rest (B)	Housing (L)	Gear (M)	Rotor and Cover (H)
201	3	2	3	15	2	0	2	14
202	1	3	7	12	2	2	1	5
203	0	11	2	21	4	3	1	1
204	2	10	14	14	4	2	2	4
205	3	6	6	30	3	3	2	14
206	4	5	7	17	4	1	5	5
102	2	2	4	27	4	1	2	19
105	1	42	9	20	3	25	29	4
107	3	8	4	27	2	3	0	12
108	3	19	34	72	2	2	2	9
110	1	8	1	49	2	13	1	42
111	3	16	4	20	3	10	3	17
112	3	12	4	13	4	4	0	8
115	3	14	3	17	3	3	0	2
116	2	2	14	21	4	1	1	12
117	1	4	6	16	2	3	3	11
121	4	13	3	33	4	7	4	20

B, baseline stage; L, low-complexity stage; M, medium-complexity stage; H, high-complexity stage.

## Data Availability

The data presented in this study are available on request from the corresponding author. The data are not publicly available due to ethical considerations.

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
