# Peer review of "Determining Cognitive Workload Using Physiological Measurements: Pupillometry and Heart-Rate Variability"

_sensors, 2024, doi:10.3390/s24062010_

Round 1

Reviewer 1 Report (New Reviewer)

Comments and Suggestions for Authors

Summary:

The authors performed an experimental study for determining cognitive workload indices in a human-centric environment using eye-tracking and electrocardiogram sensors

General concept comments: 

This is a well written paper that is scientifically sound with appropriate study design and validation. The authors have adequately explored whether physiological biomarkers can distinguish between rest, low, medium and high cognitive workloads. They have defined pupillometry and cardiac cognitive workload indices and evaluated their alignment with the task performance.

I suggest the following changes to enhance the quality of the article:

  • Please provide units for applicable metrics in Table 4

  • Please discuss limitations of the study (besides sample size) in section 4 (discussion)

  • The limitation of small sample size is discussed in Section 2.2. The authors may consider reiterating it again in section 4 from a statistical power perspective

I recommend accepting the article after the comments are satisfactorily addressed.

Author Response

Dear Reviewer:

Thank you very much for taking the time to review this manuscript.

Please find the detailed responses in the attachment and the corresponding revisions/corrections highlighted in green in the re-submitted files

We thank the reviewers for their helpful advice.

The Authors.

Reviewer 2 Report (New Reviewer)

Comments and Suggestions for Authors

1. I appreciate the authors for their good work on "Determining Cognitive Workload Using Physiological Measurements: Pupillometry and Heart Rate Variability". However, the quality of the result presentation is very bad. It should be presented better manner. For example: in figure 3 to Figure 9 - The texts are not readable. 

2. The abstract should be rewritten to convey the results. 

3. The discussion covered only 3 papers for cross-argument. There should be more papers included

4. Limitation of the work, is not written clearly  

5. Page 3, Line 143: the objective should be written clearly. I failed to understand whether it is a objective or hypothesis 

Comments on the Quality of English Language

Accept after minor revision (corrections to minor methodological errors and text editing)

Author Response

Dear Reviewer:

Thank you very much for taking the time to review this manuscript.

Please find the detailed responses in the attachment and the corresponding revisions/corrections highlighted in green in the re-submitted files

We thank the reviewers for their helpful advice.

The Authors.

Reviewer 3 Report (New Reviewer)

Comments and Suggestions for Authors

Question 1: The subject addressed in this article is worthy of investigation 

4) Agree: The topic of mental load assessment is current and actively the subject of research. A multimodal approach can provide more information in detecting mental workload. 

Question 2: The information presented is new 

4) Agree: Regarding my knowledge, assessing mental workload from metrics obtained from biological signals is a hot topic and highly interesting. Mental workload, such as stress and fatigue, are complex psychophysiological responses difficult to analyze because resulting from the interconnection of subjective, environmental, and physiological factors. A multimodal approach, such as those present in this article may be useful to have a response. 

Question 3: The conclusions are supported by the data 

5) Agree: Statistical analysis conducted in this work is thorough and accurate. I believe that the choice of tests was made with full knowledge of the facts, verifying the requirements for each test. The use of tests answers very specific specific questions and the use of a parametric or non-parametric approach must be verified precisely. The conclusions are supported by the statistical analysis. 

Question 4: The manuscript is appropriate for the journal 

4) Agree: The paper is in line with journal aim and scope.  

Question 5: Organization of the manuscript is appropriate 

4) Agree: the text is clear and readable. The important parts of signal processing are described in detail. The statistical tests and analysis were presented with a logical understanding of cause and effect. The introductory part introduces the problem and the research question. 

Question 6: Figures, tables and supplementary data are appropriate 

4) Agree: The graphs are clear and explanatory. Perhaps I would comment a little on the box plots of figures 7 and 8. Personally, the question arises to me, net of some outlayers, why for non-expert users the range of values ​​is wider than for experts, as well as the distance between first and third quartile. Perhaps greater variability in responses. Also maybe I would remove the outlayers, to dilate the Y axis and emphasize the differences. Were the two expert and non-expert samples equally numerous? But maybe that's outside the reserach question. 

Question 7: Statistical method is appropriate 

4)Agree: The statistical tests used are suitable and the approach is thorough. The very small significance level makes the test highly selective to accept or reject the null hypothesis. A single note, the use of parametric tests such as ANOVA, presumes that previous analyses have been carried out, such as the verification of normality, the equinumerosity of the compared samples and the size of the samples exceeding 23. Otherwise you should opt for a non-parametric approach. Adding this note could increase the authority of the analysis, however performed ad hoc and with full knowledge of the facts. 

In my opinion the work is well done, organized and interesting!

Author Response

Dear Reviewer:

Thank you very much for taking the time to review this manuscript.

Please find the detailed responses in the attachment and the corresponding revisions/corrections highlighted in green in the re-submitted files

We thank the reviewers for their helpful advice.

The Authors.

This manuscript is a resubmission of an earlier submission. The following is a list of the peer review reports and author responses from that submission.

Round 1

Reviewer 1 Report

Comments and Suggestions for Authors

This is an interesting application of biosensing to an applied context. However I have some concerns regarding the conclusions, as well as some other questions and comments.

First and foremost, I worry about the sample sizes used here and the conclusions that can be drawn from them. The authors collect an initial sample size of 22 participants. How was this the number they decided upon? Then, after some data exclusions they end up with only 14 participants, and only 3 of those can be classified as experts. I don't know what the standards are like for this type of work (applied ergonomics?), but I know for certain, in my field (cognitive psychology) no conclusion with a design such as this - a  between (expert vs. non-expert) x within (low, medium, high complexity) interaction - with 14 participants would be given much credence. My first suggestion to the authors is to vastly increase their sample size. The reliance on non-parametric analyses, the post hoc selection of two of the 10 dependent measures, the unclear discrepancies between the metrics, etc. all give me pause.

While I think this paper may be a good "proof of concept" demonstration regarding the use of these sensors (wearable eye-trackers and heart rate monitors), I don't think any firm conclusions regarding relative workloads across the conditions or between experts and non-experts can be made. If the authors would like to arrive at some conclusions, I would strongly recommend collecting additional data so that parametric analyses can be used and the hypotheses can be tested more rigorously.

There are several times throughout the paper that I'm not sure the results align directly with what the data are showing. For example, on lines 409-410, the authors state, "only the non-experts’ baselined PNN50 in low and high complexities are lower than the baseline." Is that true? To me, in Figure 6, it looks like the non-experts remain at/near baseline. As another example, on  lines 417-418, the authors state, "The lower HRV metrics indicating higher cognitive workload can be seen from the experts in the medium and high complexity tasks compared with the low complexity task." Just looking at Figure 6, this doesn't appear to be the case either. The line connecting the means for the experts is flat. Please be aware that any individual effect (e.g., a main effect or an interaction) not necessarily mean that pairwise comparisons are significant in the direction you hypothesized.

Second, I have some real concerns about the approach taken in the latter part of the results, shown in Figures 7 and 8. I am more familiar with pupillary dynamics, so I will focus my concerns on that. The selection of peaks and then the subsequent averaging of those peaks makes me worried about the over-estimation of effects. If your hypothesis is that cognitive workload should increase the associated pupil diameters, then you may be inflating your estimates of "increases" in pupil diameter by workload by selecting peaks. If you imaging the opposite scenario - in which you hypothesized that decreases in pupil diameter would be associated with increases in workload - you could do the opposite and select the valleys (or local minima) and average those, and perhaps find evidence for that hypothesis, too. Hopefully that illustrates my point as to why the selection and averaging of peaks is a concerning technique.

The statement in Lines 545-546, that the interaction between complexity and expertise shows an "increase in the high complexity level compared with the low or medium complexity level in either the expert or non-expert group." does not make sense and needs rewording.

Figure 4 makes it look at though the task progressed in complexity with time. But the arrow is showing increases in complexity, whereas the stages go 2-1-3. This makes it a confusing illustration.

Along those lines, why was the task constructed such that it progressed from moderately complex to low complexity to high complexity? Was there any consideration given to counterbalancing the orders across participants? At this point, the effects of complexity are perfectly confounded with time, which from an experimental point of view, is problematic.

Were the stages interdependent? That is, could someone complete rotor & cover without successfully completing gear and/or housing? 

Author Response

Dear reviewer, 

Sincerely,

the authors

Reviewer 2 Report

Comments and Suggestions for Authors

The present paper proposes a new method for measuring cognitive workload indices using heart rate variability and pupil diameter metrics, which is partially supported by empirical data. However, the report raises several questions that require further clarification and consideration.

1. Previous research has established the reasonable use of ET and ECG methods for cognitive workload investigation. This study proposes using both methods simultaneously. Thus, it is necessary to explain the relationship between these two methods, how they complement each other, and why they were chosen over other methods.

2. It is essential to clarify how potential confounding variables of participants were controlled in the study.

3. With only 14 participants, it is unclear whether the sample size is sufficient for making reliable research inferences after data processing. Therefore, additional evidence and explanation are required to support the validity of the findings.

4. The study's results show that non-experts have higher cognitive load indices in low complexities tasks than in medium complexities tasks, which contradicts previous research (such as Buchholz and Kopp (2020)) and the hypothesis. More complex tasks usually increase cognitive load, which should result in higher cognitive load indices. Why do cognitive load indices increase in low complexities tasks? It is necessary to provide a more detailed explanation for this phenomenon and investigate if there is another reason behind this observation. Additionally, the authors need to compare their results with existing research to identify similarities and differences.

Furthermore, the cognitive load index and previous method seem all unable to distinguish well between the cognitive load of low and medium complexities tasks of this study, and the same issue appears to exist in the cardiac cognitive load index method. Therefore, the researchers need to provide a more detailed and reasonable explanation for it.

5. The results using the existing method and the method proposed in this study differ in medium and high complexity tasks. It would be helpful if the authors compared the differences and characteristics of the two methods more thoroughly in the discussion section.

Author Response

Dear reviewer, 

Sincerely,

the authors

Reviewer 3 Report

Comments and Suggestions for Authors

This manuscript presents an interesting proposal to determine cognitive workload with the help of an eye-tracking and an ECG devices.

Overall, the manuscript is well presented and proposed, however I have two major issues and some minor ones. Major issues:

#1. Authors talk about expertise, but the number of 'experts' that form their sample is extremely low (3 subjects). For that reason, I think that it is not representative enough for managing differences among two groups. For that, I recommend to get rid of that comparison between groups and work with data as a unique group of 14 subjects. Another option is to acquire data of more experts and to calculate all the data and comparisons again. In case authors can justify or increment their expert group, I recommend to increase in the introduction part, the background regarding the impact that expertise has on people. There are studies in several disciplines: musicians vs non-musicians, athletes vs non-athletes, media professionals vs non-media professionals, dancers vs non-dancers, taxi drivers vs non-taxi drivers, etc. All those studies have proven together the effect on perception and task development of professional expertise. This manuscript lacks of a good background on this regard.

#2. Authors do not give the results in a clear and scientifically sound way. They give some descriptive data in, for instance, Table 3, but no Standard Deviation is presented in those data. They give p-values of comparisons among groups (experts vs non-experts) and categories of difficulty of the task, but they do not properly report the statistical analysis nor they specify what analysis was done in each case. They also talk at some moment about correlations, but do not indicate which approach was done (in discussion they talk about Pearson, but it is not the place to talk about which is the analysis developed). It is mandatory to report the statistical analysis in a clear and solid way, including each time the type of analysis done, all the data corresponding to that analysis (not only the p-value), the effect size and any particular data of specific analysis. Besides, all that has to be very well and clear presented/organized in the Results.

Other minor issues:

#1. Authors need to re-write lines 122-137 to make more clear the ideas of this parragraph.

#2. Lines 199-205: Authors need to be more clear about the final number of participants that form this study. It says 22, but then there are 14 and not specific information of what happened to the rest is provided. 

#3. Lines 216-222: Authors need to write in the same temporal time (it starts in past and finishes in future).

#4. Even English is not my principal language, I have found some minor typos/errors along the work. 

#5. I am not very sure about the justification of the complexity of the tasks. I would appreciate a clearer explanation about this.

#6. Authors need to provide information regarding the software used in the different parts of their study. For instance, in lines 304 and next, they do not talk about the software used for preprocessing the pupil behavior. Also, there is not info about the software used to analyze the data.

#7. Authors need to define every acronym used. Note SDNN, RMSS, PNN50... are not.

#8. Figure 8 y-axis indicates cognitive load values, could authors specify more about how this was obtained?

#9. Table 9 shows pupillometry and cardiac cognitive load indexes, could authors specify more about how this was obtained?

Based on all these proposals, I understand authors need to do a major revision of their manuscript, including the revision of the discussion section.

Author Response

Dear reviewer,

Sincerely,

the authors

Round 2

Reviewer 1 Report

Comments and Suggestions for Authors

I appreciate the authors' attempts to assuage my concerns with the manuscript. However many of those concerns remain. For example, the plot provided to demonstrate that peaks in pupil diameter are associated with processing tasks is not convincing to me at all. Even with the labels, it's certainly not clear that those moments are associated with peaks in pupil diameter, especially compared relatively to the large constrictions that are also occurring.

Therefore, I cannot recommend that this paper be accepted given the sample size and the types of conclusions the authors reach based on these data alone. I will defer to the editors on a decision.

Reviewer 2 Report

Comments and Suggestions for Authors

Thank you for your response. As of now, the author's revisions have made significant improvements to the paper and enhanced its quality. However, what concerns me greatly is that the study only involves three experts. Can three individuals truly represent the entire population of "experts"? The sampling error and the reliability of the results are worrisome. Although the research methodology shows innovation, it is not convincing to rely on a sample of only three individuals to represent the performance of "experts." Additionally, the mentioned effective size in the author's reply only compares the differences between two groups within the sample, which does not address the significant shortcomings in the representativeness and the substantial interference of errors caused by the three-sample approach. Most of the referenced studies utilize larger sample sizes, and the results regarding experts in this research are concerning. Therefore, I suggest that the author supplement the data with a larger expert population or use statistical methods such as robustness analysis or sensitivity analysis to demonstrate the reliability of the sample. Otherwise, the conclusions of the study appear to be unconvincing or carry significant risks.

Reviewer 3 Report

Comments and Suggestions for Authors

Dear Authors:

Many thanks for answering all of my previous comments. I add 3 more minor comments:

#1 - Table 4 now has all the information but needs to present data in a more coherent way: put the numbers with the same number of decimals, avoiding numbers raised to a power. Note that how it is now is not very practical for reading.

#2 - All the information in the Tables A-J done in the Response to Reviewer #3 comments should appear in the manuscript. I suggest that the data that is not, should be included in Supplementary materials. 

#3 - Revise the References rules of the journal. I think you are using APA, while the journal isn't. 

Author Response

Dear reviewer,

Sincerely,

The Authors
